# Promoting Authentic Academic—Community Engagement to Advance Health Equity

**DOI:** 10.3390/ijerph20042874

**Published:** 2023-02-07

**Authors:** Darrell Hudson, Keon Gilbert, Melody Goodman

**Affiliations:** 1Brown School, Washington University in St. Louis, St. Louis, MO 63130, USA; 2Behavioral Science and Health Education, St. Louis University College for Public Health and Social Justice, St. Louis, MO 63103, USA; 3Department of Biostatistics, New York University School of Global Public Health, New York, NY 10003, USA

**Keywords:** health equity, community engagement, health promotion

## Abstract

Meaningful community engagement is critical to achieving the lofty goal of health equity. Nonetheless, implementing the principles of community engagement is not easy. Attempting to implement best practices for collaborating on transdisciplinary teams and working with community partners can be challenging, particularly in locales that have a long history of strained university–community relationships. The purpose of this paper is to provide additional context and consideration for researchers, community partners, and institutions interested in conducting community-engaged research. Here, we provide guidance and highlight exemplary programs that offer effective approaches to enhance the strength of community partnerships. These partnerships not only hold promise but are also essential in the development of the local, multi-factor solutions required to address racial/ethnic inequities in health.

## 1. Background

The concept of health equity implies that everyone should be able to obtain the highest level of health possible and should not be disadvantaged because of socially determined circumstances (e.g., race/ethnicity, socioeconomic position, etc.) [1,2,3]. Community engagement has been identified as a best practice to develop approaches to address to health inequities [4,5,6]. By engaging with communities, more effective, relevant strategies and solutions can be developed to address health inequities [4,7,8,9,10]. Authentic community engagement allows for greater community buy-in, providing a seat at the decision-making table and respecting the expertise that community members possess in regard to their own lived experiences as well as their communities [11,12,13].

As community-engaged research has continually gained much-needed and deserved legitimacy, particularly over the past 30 years, academic–community partnerships have sprung up across the United States and around the globe [5,12,14,15]. There has been a rapid, substantial expansion of the community engagement literature [9,16,17]. Similarly, funding agencies, as well as organizations across different sectors, have developed guidelines on how to effectively conduct community-engaged work [14,18]. For instance, several funding agencies have developed community engagement principles, such as the Patient-Centered Outcomes Research Institute (PCORI), which includes The Engagement Rubric [19] and the National Institutes of Health Clinical and Translational Science Awards Consortium, which published the Principles of Community Engagement [20]. There are several exemplary partnerships across the country that highlight the impact of fostering collaborations to affect meaningful change, such as the Seattle-King County Healthy Homes Project [21], Harlem Children’s Zone [22], East Lake Atlanta Purpose Built Community [23], and the Fruitvale neighborhood of Oakland, California [24]. These efforts have been successful and sustainable because authentic community engagement and the intentional consideration of equity have been at the center. These efforts have also included partners across multiple disciplines and sectors, yielding more long-term impacts.

Nonetheless, there is a fine balance between the science and practice of developing and maintaining community-engaged partnerships, including but not limited to navigating power dynamics, transparency, and misalignments across institutions. Conducting effective community-engaged work is challenging as the expectations and incentive structure of academic institutions, as well as funding agencies, often collide with the priorities of community partners [25]. Community-engaged scholars have highlighted key ingredients for forging successful community partnerships, including trust, transparency, communication, commitment, and flexibility [26]. Despite the establishment and documentation of best practices that have been published in numerous texts and reports, there is no substitute for experience. Furthermore, scholars often describe community-engaged work as a value orientation and research approach rather than a specific set of methods to be followed without any adaptation [12]. As such, there are many ways that academic–community partnerships emerge and function. Considering the variance across different settings and projects, it is important to discuss how community-engaged scholars, particularly those in academic settings, and community partners collaborate, including keys to effective communication and navigating power dynamics. Yet, as is often the case in the publication process, the most positive cases are considered for publication, and some community-engaged researchers and partners may not share their stories of discord publicly [27,28,29]. These factors may obscure how practitioners address conflicts (or not) around power dynamics, communication styles, and preferences within academic–community partnerships [30].

Attempting to implement best practices for collaborating on transdisciplinary teams and working with community partners can be challenging, particularly in locales that have a long history of strained university–community relationships. The purpose of this paper is to provide additional context and consideration for researchers, particularly those that are based in academic settings, community partners, and institutions interested in promoting community-engaged research. Here, we provide guidance and highlight exemplar programs that offer effective approaches to enhancing the strength of community partnerships and that promote health equity.

## 2. Authentic Engagement

As a first step in forging a new academic–community-engaged partnership, it is critical to define what community means. Community is a broad term simply referring to a group of people, often defined by geographic area, race/ethnic groups, special interest, similar situations, or lived experience with a condition (e.g., cancer survivors or people with diabetes). Similarly, it is critical to consider which levels and sectors are most pertinent to project objectives. Once the most appropriate partners are at the table, it is important that all stakeholders are mindful of their own experiences and positionality, including identity-related influences and professional biases. Transparency regarding the positionality of team members aids in the development of mutual trust and understanding. For example, among researchers based in academic settings, adhering to project timelines is incredibly important for grant-funded research and maintaining relationships and trust with funders. However, community partners may operate within organizations that have different norms regarding time and organizational priorities [31]. Similarly, it is critical to provide space for collaborators to share where they are regarding bandwidth, time, and energy. Communicating about the expectations, norms, and stressors that collaborators are contending with builds trust and empathy, in addition to helping set expectations and procedures about completing work, especially when there are time constraints [31].

In general, community-engaged projects often require additional time and need to be reflexive of community input. For university-based researchers, there is often pressure, both from funders and university directives, to prioritize the adherence to project timelines and deliverables over being reflexive or incorporating meaningful feedback from community partners. To engage with community partners in an authentic manner, community-engaged scholars must be flexible in order to incorporate the perspectives of community members [26,32]. Furthermore, conflicts, whether minor or major, are natural in collaborative work [31]. Conflicts may emerge from a wide range of differences related to positionality, personality, workstyle, and more. It is helpful to define common values, such as empathy and transparency, and to establish ground rules for teams to define how to handle conflicts proactively. These include strategies such as listening to understand and finding ways to reach common ground. As conflicts emerge within teams, it is helpful to revisit the shared agreements, values, and ground rules. Open lines of communication can help improve understanding among team members, especially how they can best support each other and, if necessary, adjust expectations in order to conduct and produce the best work [31].

Another key challenge related to authentic engagement is sharing power. The first step is to engage in an honest discussion about perceptions of power differentials on a team [31]. These discussions may help team members recognize the complementary strengths that partners have. Taking inventory of these collective and complementary strengths helps team members see how they can support one another. These discussions are also enhanced by finding ways to build trust, such as finding commonalities and shared experiences [33]. Similarly, it is important to identify common values and develop guiding principles that will help determine decision-making within the partnership. These values and principles are helpful to reference when differences in opinion emerge or when there is a disagreement about what direction to take when there are conflicting opinions [31]. These discussions and team-building activities take time. If there is not adequate time for authentic dialogue between community partners and research team members, transparency and trust can break down, threatening the sustainability of partnerships. Similarly, it is critical to allow time for feedback on the process and deliverables that any team proposes. This process may even need be prioritized over markers of project success. 

If the feedback of community partners is not incorporated, the viability and long-term sustainability of a project will come into question. Goodman and Sanders Thompson described three broad levels of stakeholder engagement, namely non-participation (e.g., outreach and education), symbolic participation (e.g., coordination and cooperation), and engaged participation (collaboration, patient-centered, and participatory) [4]. It is critical for community-engaged scholars and their partners to consider which level is most appropriate and whether to set up a partnership for the long term or the development of a singular project. This determines the expectations for engagement, including roles, frequency, and the long-term sustainability of a project. From a community perspective, authentic engagement means their voice is heard, their suggestions are being incorporated and implemented, they are involved in decision- making and have shared power with academic researchers, and the outcomes of the partnership have some direct benefit to them, their organization, and/or their community [33].

## 3. Institutional Barriers to Community Engagement

Decisions regarding which level of engagement scholars choose and how long partnerships should last do not occur in a vacuum [31,33]. For example, funders and universities play a critical role in determining the shape and longevity of community–academic partnerships. It is critical for these institutions to evaluate whether their priorities are informed by and appropriately aligned with the interests of their community partners. Additionally, institutions must be intentional in how to best support scholars, especially those from historically underrepresented backgrounds and those that are pre-tenure.

One key institutional challenge is accounting for the active partnerships that university faculty and staff are maintaining. While it is seemingly impossible to keep track of all activity, community organizations and leaders often feel overwhelmed by the requests of researchers to partners. Emerging funding announcements often emphasize community engagement, and there is a greater impetus for academic institutions to engage in this kind of work with external incentives. With so many requests from potential academic partners, community organizations and members can be overwhelmed. This is especially true if researchers are not allotting enough time to develop true partnerships, limiting opportunities to develop relationships and establish ground rules. Greater coordination on the part of universities would be helpful in making sure that potential community partners are not inundated with requests to collaborate. It is also important to note the power differentials within academic–community partnerships. For example, even if a request from a local university is not aligned with the mission or capacity of an organization, community partners may feel that they must participate in order to avoid being excluded from opportunities in the future.

A related issue is the temptation of academic researchers to overpromise when working with community members or funders. However, there should be a straightforward manner of communicating expectations and decisions among team members. Simultaneously, it is important to consider ways to ensure that the goals of the project do not overstretch the expertise/capacity of the team and the individuals that comprise the team. For example, there have been several seismic events, including but not limited to the police-involved slayings of unarmed Black people, including Michael Brown, Breonna Taylor, and George Floyd [34,35,36]. In addition, there have been attacks on Black Americans in safe spaces such as grocery stores and churches by White supremacists [37]. These events have occurred in addition to the COVID-19 pandemic as well as the social, economic, and outcome disparities related to COVID-19. These events have underscored inequities embedded into the fabric of the United States [38,39,40]. In the wake of these tragedies, there has been a desire to make changes to address inequities and increased pressure for institutions to develop solutions, especially from universities [38,40]. While there is pressure for institutions to respond, there must be systematic efforts to develop innovative approaches to address health inequities and to meaningfully engage communities. Without these commitments, community-engaged efforts are merely an academic exercise with minimal impact on communities that have long been reeling from systemic marginalization and brutality. Additionally, it is important to recognize the potential impact of these national events on the well-being of community partners, especially those who are people of color, as scholars have noted that vicarious racism, described as indirect experiences of racism that may play out on television or social media, is a threat to mental health and overall wellbeing [41,42,43].

## 4. Fertile Ground: How Academic Institutions Support Community-Engaged Work

Establishing training and capacity building within academic institutions is helpful in aiding scholars to better and more effectively engage community partners. Community partners should also be involved in training, particularly in the navigation of perceived differences in power [44,45]. As outlined below, it is critical to train researchers in best practices related to community engagement and to help community partners develop skills that foster their ability to engage in processes and develop solutions to challenges that they want to address. These programs also foster confidence in partners’ ability to bring their perspectives to the table and better manage relationships. There are exemplars of academic–community partnerships as well as funding opportunities that build capacity and foster collaboration. For example, the Community Research Fellows Training (CRFT) program helps to build the capacity of community members to enhance their understanding of research, key factors related to health equity, and to build a base of knowledge related to the development of solutions for health inequities at a local level [11,16,45]. The program has endeavored to develop equitable and mutually beneficial partnerships. This training program included the delivery of content in a 15-weekly, three-hour training format as well as seed funding that community fellows can use to develop their own projects [44,45]. Over the 10 years since its inception, CRFT has received funding from a variety of sources. In St. Louis, the CRFT program and the work of the project team were sponsored by the Program to Eliminate Cancer Disparities (National Institutes of Health, National Cancer Institute grant U54CA153460), the Siteman Cancer Center, Staenberg Family Foundation, and the Barnes-Jewish Hospital Foundation. In Mississippi, the state department of health, a grant from the state, and institutional support from the University of Southern Mississippi. In New York City, CRFT is funded by New York University School of Global Public Health.

There has also been a great deal of organic cross-pollination of fellows who were drawn from different sectors and experiences. This aids in sustainability as fellows develop skills and connections that go beyond the structure of the training program. The impact of these connections is difficult to quantify, but fellows have maintained connections and incorporated the materials they were exposed to in training into their positions [16]. Evaluation data have indicated that the program is successful in increasing trainees’ knowledge of social determinants of health and the research process. Furthermore, the program has been successfully implemented in several locales and different universities, including Stony Brook University, Washington University in St. Louis, University of Southern Mississippi, Mississippi College, and New York University.

Another example of a funding opportunity that prioritizes the development of sustainable, effective partnerships is the Interdisciplinary Research Leaders (IRL) program, a Robert Wood Johnson Foundation initiative [33]. This three-year program provides funding to fellows, often comprised of academic researchers and community partners, to develop leadership skills that span different sectors and disciplines, creating a national network of leaders to advance health equity. IRL seeks to foster action-oriented research that is beneficial to local communities and is ethical, assessable, and aligned with community standards. For researchers, this training enhances their ability to authentically engage with community partners, translate research into policy and practice, and understand how the policy process works at different levels. IRL builds the capacity of community partners, enhancing their ability to engage with researchers as well as how to leverage research findings to advocate for policy changes. The program provides funding for a research project in which there is an intentional focus on the development of a sustainable partnership. This training is delivered through frequent webinars and national meetings that bring together teams from all over the U.S. to share their work collaborative leadership process and to network in addition to content that is provided asynchronously through a learning management system [33]. These efforts not only assist in the development of community capacity to more fully participate in partnerships, they make the playing field more even, centering power rather than allowing academic researchers to drive directions and activities [31,33].

## 5. Conclusions

The African Proverb “If you want to go fast, go alone. If you want to go far, go together” is an appropriate way to summarize academic–community partnerships. As community-engaged work continues to grow, the art, science, and practice of developing effective partnerships will evolve, and more solutions to addressing health inequities will emerge [11]. Community-engaged scholars should anticipate that there will be conflicts and proactively devise approaches to handle disagreements in a respectful, professional manner that will not undermine feelings of trust or respect among team members [31,33].

As part of establishing mutual respect and trust among team members and with community partners, it is essential to share power and decision-making. Therefore, it is critical for scholars genuinely interested in working collaboratively on teams and/or engaging with communities to enter the work ready to share power. This means being transparent and clear in the process of making decisions, as well as allowing adequate time and space to discuss issues and finding ways to engage with people who have different opinions, perspectives, and/or work habits. Additionally, community-engaged scholars should be sure to build in enough time to solicit community perspectives and augment their action plans according to feedback garnered from community members and organizations.

The people who are often excluded from this work, those that are “hard to reach” using existing recruitment and engagement approaches and incorporate into the work, are people that might have the most insightful and compelling perspectives. Community-engaged scholars must also recognize that incorporating only “grass top” organizations and individuals will not yield representative or adequately inclusive representation from the key “grass roots” stakeholders required to affect change. Despite structural challenges, such as racial residential segregation, people who have been historically marginalized are remarkably resilient and are experts in their own lived experiences. They must be respected and brought to the table if progress is to be made. 

Authentic community engagement is an effective pathway to developing solutions that can address health inequities [4,46]. Community members are experts in their own lived experiences, and it is important to engage with them to identify the most salient factors to foster the resilience many people are already displaying in challenging circumstances [14,18]. Previous efforts have indicated that community organizing is impactful, stimulating the local economy and engaging local elected officials to address issues related to health inequities [47]. The exemplar programs discussed in this paper shed light on efforts that have successfully built community capacity as well as the bidirectional communication and leadership approaches needed to forge authentic, impactful engagement efforts.

## Data Availability

Not Applicable.

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
