# Peer review of "Promoting Authentic Academic—Community Engagement to Advance Health Equity"

_ijerph, 2023, doi:10.3390/ijerph20042874_

Round 1

Reviewer 1 Report

Community engaged research continues to be an important topic and there is plethora of emerging research to support importance of community engaged research in advancing health equity.  Overall, the manuscript descriptively describes the issues relating to foster community engaged collaborations and the challenges. The literature I believe has covered this extensively. The manuscript would make more of contribution if it clearly and systematically links the community engaged framework to advancing health equity. What are the important components of community engaged collaborations that move the needle toward health equity? What are specific strategies needed? Are their exemplars that specific illustrate this? The authors could expand on this more. 

Author Response

Community engaged research continues to be an important topic and there is plethora of emerging research to support importance of community engaged research in advancing health equity.  Overall, the manuscript descriptively describes the issues relating to foster community engaged collaborations and the challenges. The literature I believe has covered this extensively.

We appreciate the comment but respectfully disagree. As mentioned in the manuscript, there has been tremendous growth in the literature and lots of discussion of these issues anecdotally. However, there are few papers that provide details around the challenges described here such as coordinating efforts or identifying appropriate community partners. Nor are there many examples in the existing literature of how researchers and academic institutions can help build community capacity.

The manuscript would make more of contribution if it clearly and systematically links the community engaged framework to advancing health equity.

We believe we have made this link explicit in the manuscript:

Community engagement has been identified as a best practice to addressing health inequalities [1]–[3]. By engaging with communities, more effective, relevant strategies and solutions can be developed to address health issues. Authentic community engagement allows for greater community buy-in, providing a seat at the decision-making table and respecting the expertise that community members possess in regard to their own lived experiences as well as their communities [4]–[6].

What are the important components of community engaged collaborations that move the needle toward health equity? What are specific strategies needed? Are their exemplars that specific illustrate this? The authors could expand on this more.

The exemplars discussed in the paper shed light on programs that have successfully built community capacity as well as the bidirectional communication and leadership approaches needed to forge authentic, impactful engagement efforts. We have reiterated this contribution in the conclusions section.

Reviewer 2 Report

Overall, this manuscript adds value to the field. The background provides a good rationale, and the conclusion anchors the paper’s core message. In general, the authors should review when and when they are not using hyphens for ‘community engaged’ throughout manuscript. Several inconsistencies are observed. As the paper focuses on authenticity, terminology such as positionality and community interests would add value to this paper. Also, please confirm that NOT indenting the first paragraph of a new section is aligned with the journal’s format.

Reviewer comments are segmented by SECTION.

BACKGROUND

-          Include references to support the concept of health equity.

-          Give a description of what community can mean in the context of community engagement.

-          The following lines have placeholders for citations that have not been filled in:

o   Line 35

o   Line 49

-          Line 32: delete space that is between hyphen for community-engaged.

-          Lines 35-37: authors state funding agencies and organizations across different sectors, but do not demonstrate this. Only federal agencies are acknowledged. Authors should describe another sector where organizations or funding is devoted to community-engaged research.

-          Line 41: change partnership to partnerships.

-          Several partnerships are mentioned in lines 42-43. It is not clear if they are partnerships between community and the academy or some other form of collaboration. Please provide a sentence or two that provides some context to why/how these are exemplar partnerships. There is a concern that readers may simply draw that they are outstanding partnerships because they simply focus on neighborhoods characterized by communities of color. This could encourage health equity tourism.

-          Lines 44-45: revise this sentence. ‘community-engaged researcher participation and partnership’ does not read well.

-          Lines 49-51: within these lines, the sentence draws questions:

o   Who considers community-engaged work this way?

o   Is it bad to view community-engaged work this way?

o   How do the authors want researchers and funding agencies to view this work?  

-          Line 53: ‘conflicts within partnership’ is not clear. Please spell this out a little more so that readers do not have to make assumptions about the context of within partnerships.

-          Line 58: The authors do not talk about how community-engaged research promotes health equity. Rather, this manuscript provides guidance on how to be equitable and authentic in partnerships/collaborations. Given the chosen title and this sentence, authors may need to add a paragraph to this piece that connects the dots between community engagement and advancing health equity (either as a process or outcome) for the audience.

AUTHENTIC ENGAGEMENT

-          This section could be strengthened by adding a few sentences that communicate what authentic engagement is or looks like to community partners. This section mainly centers authentic engagement for researchers.

-          Line 71: need to follow with citation after naming authors. It would be helpful if authors share the implications for not being clear on the level of stakeholder engagement for (potential) community partners.

INSTITUTIONAL BARRIERS

-          The order and number of barriers discussed are not clear. Transition words do not provide the needed context clues in paragraph 1 of this section. Further, paragraph 2 clearly calls out another institutional challenge, but the reader has not been sufficiently told what the first institutional challenge was.

-          Line 78: missing a word or phrase between who and historically.

-          Line 87: first mention of power. This manuscript would be improved with more presentation of perspective on power differentials and how that must be disrupted to advance community-engaged research/partnership and health equity. An important question this manuscript does not answer is where or how should power be placed in community-academic partnerships? This is important guidance to offer readers. Sharing power is also mentioned in conclusion, but it is more at the interpersonal level and not written in a way that communicates how it shifts institutional relationships that benefit community partners.

-          Lines 93-96: The reviewer encourages authors to be clearer in communicating their intention/purpose behind focusing on one racialized group and no other systematically disadvantaged groups here. It is important to be upfront about this (not to justify, but to convey the author’s narrative)

o   In support of centering Black folks, the reviewer strongly encourages the authors to provide some citations about these national events and calling out the perpetrators of these acts. If the authors choose to continue their focus solely on Black people, the word/phrase ‘attacks’ used in line 95 could be stronger.

-          The sentence starting on Line 96 is too vague to stand alone, without accompanying citations.

FERTILE GROUND

-          Line 106-107: This sentence should also speak to power in some form.

-          Please include some mention of the sponsor of CRFT that is similar to what was done for IRL.

-          Line 117: Need a citation for evaluation data.

-          Line 118: change locals to locales.

CONCLUSIONS

-          Line 133: revise this sentence. There is a typo or incorrect phrase.

-          Lines 146-150: This paragraph seems to depart, in perspective, from the paper’s core. Consider making this paragraph about health equity tourism rather than tokenism. I think this will ring truer to the challenge of authentic community engagement and health equity.

-          Line 151: Authors are strongly encouraged to not use the phrase “hardest to reach”. Based on the examples used in this manuscript, this carries a connotation that Black communities are hardest to reach, but does not explicitly name the biases, barriers, and systems that sustain the issues between Black communities and academic institutions.

Author Response

We appreciate the reviewer's helpful comments. We have attached our responses to each of your concerns. 

Round 2

Reviewer 2 Report

This paper contains powerful prose that support improving academic-community partnerships to advance health equity. The following are minor edits: 

- line 67, As should be lowercase. 

- line 70, remove the word which.

- line 80, I suggest saying academic-community partnerships instead of community partnerships. 

- line 82-100, ensure this is one paragraph. 

- line 192, remove the word as. 

- line 286, include a comma (,) after the word work. 

Author Response

We appreciate the reviewer's careful examination of the manuscript. We have addressed each of the errors that were highlighted in this drat.